# Impact of In Vitro Digestion on the Digestibility, Amino Acid Release, and Antioxidant Activity of Amaranth (*Amaranthus cruentus* L.) and Cañihua (*Chenopodium pallidicaule* Aellen) Proteins in Caco-2 and HepG2 Cells

**DOI:** 10.3390/antiox12122075

**Published:** 2023-12-05

**Authors:** Gloricel Serena-Romero, Anaís Ignot-Gutiérrez, Osvaldo Conde-Rivas, Marlenne Y. Lima-Silva, Armando J. Martínez, Daniel Guajardo-Flores, Elvia Cruz-Huerta

**Affiliations:** 1Centro de Investigaciones Biomédicas, Universidad Veracruzana, Dr. Luis Castelazo Ayala s/n, Industrial Ánimas, Xalapa 91193, Veracruz, Mexico; 2Instituto de Neuroetología, Universidad Veracruzana, Dr. Luis Castelazo Ayala s/n, Industrial Ánimas, Xalapa 91193, Veracruz, Mexico; 3Facultad de Nutrición-Xalapa, Médicos y Odontólogos s/n, Unidad del Bosque, Xalapa 91017, Veracruz, Mexico; 4Tecnológico de Monterrey, Escuela de Ingeniería y Ciencias, Centro de Biotecnología FEMSA, Eugenio Garza Sada 2501 Sur, Monterrey 64849, Nuevo León, Mexico; 5Centro de Investigación y Desarrollo en Alimentos, Universidad Veracruzana, Dr. Luis Castelazo Ayala s/n, Industrial Ánimas, Xalapa 91193, Veracruz, Mexico

**Keywords:** amaranth, cañihua, proteins, simulated gastrointestinal digestion, antioxidant activity, bioactive peptides

## Abstract

This study evaluated the impact of in vitro gastrointestinal digestion on the digestibility, amino acid release, and antioxidant activity of proteins from amaranth (*Amarantus cruentus* L.) and cañihua (*Chenopodium pallidicaule* Aellen). Antioxidant activity was assessed using ORAC, ABTS, DPPH, and cellular antioxidant activity (CAA) assays in human intestinal Caco-2 and hepatic Hep-G2 cell lines. The results showed that amaranth had higher protein digestibility (79.19%) than cañihua (71.22%). In addition, intestinal digestion promoted the release of essential amino acids, such as leucine, lysine, and phenylalanine, in both protein concentrates. Concentrations of amaranth and cañihua proteins, ranging from 0.125 to 1.0 mg mL^−1^, were non-cytotoxic in both cell lines. At a concentration of 0.750 mg mL^−1^, simulated gastrointestinal digestion enhanced cellular antioxidant activity. Intestinal digest fractions containing peptides >5 kDa were the principal contributors to CAA in both cell lines. Notably, cañihua proteins exhibited high CAA, reaching values of 85.55% and 82.57% in Caco-2 and HepG2 cells, respectively, compared to amaranth proteins, which reached 84.68% in Caco-2 and 81.06% in HepG2 cells. In conclusion, both amaranth and cañihua proteins, after simulated gastrointestinal digestion, showcased high digestibility and released peptides and amino acids with potent antioxidant properties, underscoring their potential health benefits.

## 1. Introduction

Protein digestibility is critical in determining proteins’ nutritional value and utilization. It refers to how proteins are broken down and absorbed in the gastrointestinal tract. Proteins with high digestibility provide a rich source of bioavailable amino acids, essential for various physiological processes, including the growth, repair, and maintenance of body tissues [1]. During gastrointestinal digestion, proteins undergo enzymatic hydrolysis, releasing peptides and free amino acids. These peptides and amino acids play a crucial role in human health as they possess bioactive properties that can influence physiological functions [2]. Among these bioactive properties, the antioxidant activity of peptides and amino acids has gained important attention.

Antioxidants play a vital function in counteracting the harmful effects of free radicals. They function through mechanisms such as hydrogen atom transfer or inhibition of electron migration, effectively neutralizing free radicals [3,4]. Consuming a healthy diet rich in bioactive compounds that act as natural antioxidants has been shown to reduce the risk of various pathologies [5], for example, liver diseases [6] and cardiovascular [7], neurological [8], and renal pathologies [9,10].

Among these compounds, bioactive peptides have gained considerable attention due to their ability to exert beneficial effects on human health, including antioxidant properties. The antioxidant properties of bioactive peptides are influenced by their sequence, structure, and amino acid composition [11,12]. Histidine has been described to donate hydrogen atoms and inhibit peroxide radicals, while aromatic amino acids such as phenylalanine, tryptophan, and tyrosine donate hydrogen atoms and neutralize free radicals [13]. Also, cysteine, with its sulfhydryl (SH) group, can scavenge free radicals [14]. Hydrophobic residues, including H, W, F, P, G, K, I, and V, not only contribute to the antioxidant activity of peptides but also enable their interaction with lipid bilayer membranes and reach target organs [3,15,16], and peptide solubility is also crucial as hydrophobic residues tend to donate protons to radicals. Consequently, histidine residues with higher hydrophobicity, often found in the N-terminal region of peptides, show greater accessibility to peroxyl radicals due to the presence of acyl groups [17].

In this sense, understanding the bioactivity and antioxidant potential of peptides and amino acids derived from the digestion of pseudocereal proteins such as amaranth and cañihua is of great importance in food and nutrition. Also, two species of this group such as amaranth *A. cruentus* L. and cañihua *Ch. pallidicaule* have gained recognition for their outstanding nutritional value [18].

These grains are characterized by their high protein content (approximately 13.6% and 15.2%, respectively, on a dry weight basis) and composition of essential amino acids [19]. In in vitro studies, amaranth has been associated with various beneficial health effects, including antidiabetic [20], anticancer [21], and antioxidant activities [14]. Likewise, cañihua proteins have demonstrated antimicrobial, antihypertensive, and antioxidant activities [22,23,24]. The evaluation of their potential health benefits can lead to developing functional foods and dietary strategies to promote their consumption and prevent chronic diseases. Therefore, the aim of this study is to investigate the release of amino acids during the in vitro digestion of amaranth and cañihua proteins and their antioxidant activity. 

## 2. Materials and Methods 

### 2.1. Materials and Chemicals

Pepsin from porcine gastric mucosa (Sigma-Aldrich, P7012), pancreatin from porcine pancreas (Sigma-Aldrich, P7545), bovine bile (Sigma-Aldrich, B3883), 4-(2-aminoethyl) benzenesulfonyl fluoride (Pefabloc^®^ SC, Sigma-Aldrich, 76307), ninhydrin, dichloro-dihydro-fluorescein diacetate (DCFH-DA), 2,2′-azobis (2-methylpropionamide)-dihydrochloride (AAPH), 6-Hydroxy-2,5,7,8-tetramethylchroman-2-carboxylic acid (Trolox), 2,2′-azino-bis(3-etilbenzotiazolina-6-sulfónico) (ABTS), and 2,2-Diphenyl-1-picrylhydrazyl (DPPH) were from Sigma-Aldrich (Milwaukee, WI, USA). Sigma-Aldrich Dulbecco’s Modified Eagle Medium (DMEM-F12) was obtained from Thermo Fischer Scientific (Waltham, MA, USA). Fetal bovine serum (FBS), phosphate-buffered saline pH 7.4 (PBS), trypsin-EDTA 0.25%, penicillin (10,000 Unit mL^−1^), and streptomycin (10,000 μg mL^−1^) were acquired from GIBCO (Grand Island, NY, USA). CellTiter 96 ^®^ Aqueous One Solution Cell Proliferation Assay was obtained from Promega Corporation (Madison, WI, USA). The rest of the chemicals used were of HPLC grade. All chemicals were from Sigma-Aldrich, Merck (Saint Louis, MO, USA), unless otherwise specified in the respective section.

### 2.2. Plant Material 

Amaranth seeds (*Amaranthus cruentus* L.) were obtained directly from growers in Puebla, Mexico, while cañihua seeds (*Chenopodium pallidicaule* Aellen) were obtained from Food to Live in New York, NY, USA. Each seed type was individually processed using a pulverizing mill (ENCAMEX, MC-100, Cd Mexico, Mexico) and then sieved through a mesh number 60. The obtained flours were stored under vacuum at −80 °C until required for analysis.

### 2.3. Protein Extraction and Quantification

Protein extraction was carried out using the isoelectric precipitation method previously described by Chirinos et al. [25], with slight modifications. Defatted flours of both amaranth and cañihua were individually suspended in water (1:10 *w*/*v*). The pH was adjusted to 10 using 1 M NaOH. These suspensions were stirred at 200 rpm, maintained at 50 °C for 60 min, and filtered. After filtration, samples were centrifuged at 10,000× *g* for 20 min at 4 °C. The pH of supernatants was adjusted to 4.8 with 5 M HCl, followed by another centrifugation round at 10,000× *g* for 10 min at 4 °C. The collected precipitates were rinsed twice, re-suspended in water, and neutralized using 1 M NaOH. These suspensions were then freeze-dried and stored at −80 °C for subsequent analyses. For the proximate composition assessment of the uniform flour samples, standard AOAC methods [18] were employed. Moisture (934.01), fat (930.09), ash (930.05), crude fiber (934.10), and protein (978.04) contents were determined. The protein content was calculated using a conversion factor of 6.25. The protein content in the amaranth and cañihua protein concentrates was determined by using the Kjeldahl method. 

### 2.4. In Vitro Gastrointestinal Digestion 

Protein concentrates were digested following the harmonized protocol of static digestion INFOGEST [26]. Initially, the protein concentrate was mixed in a ratio 1:1 (*w*/*v*) with simulated salivary fluid (SSF) at pH 7 containing salivary amylase (75 U mL^−1^). This mixture was incubated at 37 °C with an agitation of 150 rpm for 2 min, maintaining a pH of 7. Following the oral phase, the mixture was combined at a ratio 1:1 (*v*/*v*) with simulated gastric fluid (SGF) at pH 3, which included pepsin (2000 U mL^−1^) and was incubated at 37 °C, with agitation, maintaining a pH of 3 for 2 h. Finally, to simulate the intestinal phase, the remaining digest was added in a ratio 1:1 (*v*/*v*) with simulated intestinal fluid (SIF) at pH 7, bile (10 mM), and pancreatin (100 U mL^−1^) and was incubated again with agitation at 37 °C and an adjusted pH at 7 for 2 h. To stop the activity of digestive enzymes, Pefabloc^®^ was added. Subsequently, the samples were freeze-dried and stored at −80 °C for subsequent analyses. The specific enzymatic activities, ratios, and concentrations of the materials in the digestion simulation strictly followed the referenced protocol. Digestion was performed in duplicate. A control digestion blank was also prepared, containing the enzyme mixture at the corresponding concentration without protein concentrate. Both the gastric and intestinal digests, after 120 min, were ultrafiltered through a 5000 Da cutoff membrane (Vivaspin 2, 5 kDa MWCO, Cytiva Life Sciences; Marlborough, MA, USA). The resultant fractions < 5 kDa and >5 kDa were freeze-dried and stored at −80 °C until use. The protein content in the digests and the fractions was determined by using the bicinchoninic acid method (BCA) (Pierce, Rockford, IL, USA), using bovine serum albumin as the standard protein.

### 2.5. In Vitro Protein Digestibility 

Amaranth and cañihua protein concentrates were subjected to in vitro digestion using the harmonized protocol of static digestion of INFOGEST previously described [26]. Samples of digested proteins were collected at various time intervals: 0, 30, 60, 90, 120, 150, 180, 210, and 240 min throughout the gastric and intestinal digestion phases. Collected samples were centrifuged at 5000× *g* at 4 °C for 30 min, and the supernatant was freeze-dried and stored at −80 °C until analysis. The Bradford method was used to evaluate the protein content using bovine serum albumin (BSA) as a standard protein. The in vitro digestibility of the proteins from amaranth and cañihua was measured according to the equation described by Babatunde et al. [27]:(1)$$Digestibility\;\left(\%\right)=\frac{\left(I-F\right)}{I}\times 100$$
where *I* represents the protein content of the sample before in vitro digestion, and *F* is the protein content of the sample after in vitro digestion.

### 2.6. Sodium Dodecyl Sulfate-Polyacrylamide Gel Electrophoresis (SDS-PAGE)

Samples were dissolved at a concentration of 1 mg mL^−1^ in 10 mM Tris-HCl buffer containing 2% (*w*/*v*) SDS and 5% (*v*/*v*) 2-β-mercaptoethanol and heated at 95 °C for 4 min. Subsequently, 10 µL of the samples was loaded onto 12% bis-tris polyacrylamide gels using a Mini-PROTEAN^®^ Tetra cell (Bio-Rad Laboratories Inc., Hercules, CA, USA). A molecular weight marker (Precision Plus Protein™ Unstainend standard, Bio-Rad) was used as a reference. Electrophoretic separation was performed for 60 min at 150 V using XT MES running buffer (Bio-Rad) and a PowerPac™ Basic Power Supply (Bio-Rad). After completion of electrophoresis, the gels were carefully removed from the cassettes, rinsed twice with distilled water, and stained with Coomassie blue solution (50% water, 40% methanol, 10% Coomassie blue) for 1 h. The gels were then rinsed with distilled water before adding the de-stain solution (70% water, 20% methanol, 10% glacial acetic acid) overnight with gentle shaking. Finally, the visualization and analysis of the protein bands were performed using the GelAnalyzer 19.1 software.

### 2.7. Free Amino Acid Analysis

The free amino acids in the digested samples were determined using the methodology described by [28], with minor modifications. After lyophilizing the digests, 0.2 g was dissolved in 4 mL of 5-sulfosalicylic acid (5%) and agitated for 1 min. It was then stored at 4 °C for one hour and centrifuged at 15,000× *g* for 15 min at the same temperature. The supernatant pH was adjusted to 2.2 using 0.3 M NaOH and filtration through a 0.45 μm membrane. Free amino acids were assessed using cation exchange ion chromatography with post-column derivatization with ninhydrin. Concentrations were determined colorimetrically at 440 and 570 nm using an amino acid analyzer (JLC-500/V AminoTac™, MA, USA).

### 2.8. Antioxidant Activity 

#### 2.8.1. Oxygen Radical Absorbance Capacity (ORAC)

The ORAC-fluorescein (ORAC-FL) assays were conducted following the method described by Hernández-Ledesma et al. [29] with some modifications. The reaction mixture consisted of a phosphate buffer (75 mM) at pH 7.4 and was incubated at 37 °C. A mixture of 200 μL was prepared to contain FL acid (70 nM), 2,2′-azobis(2-methylpropionamide)-dihydrochloride (AAPH, 14 mM), and either antioxidant acid [6-hydroxy-2,5,7,8-tetramethylchroman-2-carboxylic acid] (Trolox, 10–60 µM) or the sample at various concentrations. Fluorescence measurements were taken for 1 h at 2-min intervals using a microplate reader (BioTek^®^ Synergy HT KC4, Winooski, VT, USA) with a 485 nm excitation filter and a 520 nm emission filter. The assay was performed in triplicate, and the results were expressed as micromoles of Trolox equivalents per gram of sample on a dry weight basis (μmol TE mg^−1^). 

#### 2.8.2. ABTS Radical Scavenging Assay

The ABTS radical scavenging activity assay was performed based on the method described by Re et al. [30] with some modifications. The monocation radical ABTS^•+^ was generated by reacting ABTS (7 mM) with potassium persulfate (2.45 mM) at room temperature (25 °C) for 16 h in the dark. The ABTS^•+^ radical solution was then diluted with phosphate buffer at pH 7.4 to obtain an absorbance value of 0.70 (±0.1) at 734 nm. In a 96-well plate, 20 μL of the sample and 180 μL of the ABTS^•+^ radical solution were mixed and allowed to react for 5 min. The absorbance was then measured at 734 nm using a microplate reader (Multiskan FC model IVD, Vantaa, Finland). A Trolox calibration curve was prepared, and the Trolox equivalent antioxidant capacity (TEAC) value was expressed as micromoles of Trolox equivalents per milligram of the sample (μmol TE mg^−1^).

#### 2.8.3. DPPH Radical Scavenging Assay

The DPPH radical scavenging activity assay was conducted following the procedure outlined by Brand-Williams et al. [31] with slight modifications. A solution of 2,2-diphenyl-1-picryl-hydrazyl-hydrate (DPPH) at a concentration of 100 µM in 80% methanol was prepared. In a 96-well plate, 20 µL of the sample and 180 µL of the DPPH solution were combined and incubated for 30 min in the absence of light. The absorbance was then measured at 517 nm using a microplate reader (Multiskan FC model IVD, Vantaa, Finland). A Trolox calibration curve (10–100 µM) was performed. The results were expressed as micromoles of Trolox equivalents per gram of sample (μmol TE mg^−1^).

#### 2.8.4. Cell Culture 

The human colorectal adenocarcinoma cell line (Caco-2) and hepatocarcinoma cell line (HepG2) were obtained from the American Type Culture Collection (ATCC, Rockville, MD, USA). The cells were cultured in Dulbecco’s Modified Eagle’s Medium (DMEM, Sigma Chemical) supplemented with 10% (*v*/*v*) fetal bovine serum (FBS, BioWest, Nuaillé, France), 1% (*v*/*v*) penicillin/streptomycin/amphotericin B solution (BioWest), and 1% (*v*/*v*) non-essential amino acids (Lonza, Walkersville, MD, USA). Cell cultivation conditions were set at 37 °C in a humidified atmosphere with 5% CO_2_. The culture medium was refreshed every two days, and cell subconfluency was maintained using trypsin/EDTA (Lonza Group Ltd.). 

#### 2.8.5. Cell Viability Assay

Cell viability was evaluated using the MTS CellTiter 96^®^ AQueous One Solution Cell Proliferation Assay. Caco-2 and HepG2 cells were seeded onto 96-well plates at a density of 6 × 10^4^ cells/well in a complete medium enriched with 10% FBS and subsequently incubated at 37 °C for 24 h. Post incubation, the culture medium was discarded, and the cells were treated with the sample solutions with concentrations of 0.250, 0.750, and 1 mg mL^−1^ for an additional 24 h. After that, the cells were washed with phosphate-buffered saline (PBS, Lonza Group Ltd.). The assay involved adding 20 µL of the CellTiter 96^®^ AQueous One Solution Reagent, comprising the tetrazolium compound [3-(4,5-dimethylthiazol-2-yl)-5-(3-carboxymethoxyphenyl)-2-(4-sulfophenyl)-2H-tetrazolium, inner salt; MTS] and phenazine ethosulfate (PES), directly to the culture wells. After 1 h incubation, the absorbance was measured at 490 nm using a BioTek^®^ Synergy HT KC4 microplate reader (Winooski, VT, USA). All experiments were conducted in triplicate, and results were expressed as the percentage of viable cells relative to the control (untreated cells), considered 100%.

#### 2.8.6. Cellular Antioxidant Activity

Cellular antioxidant activity (CAA) was assessed using a method adapted by Gutiérrez-Grijalva et al. [32]. Human colorectal adenocarcinoma (Caco-2) and hepatocarcinoma (HepG2) cells were individually cultured on a black 96-well plate until reaching a density of 5 × 10^4^ cells/well. These cells were maintained in DMEM supplemented with 10% fetal bovine serum at 37 °C in a 5% CO_2_ humidified atmosphere. After achieving the desired density, cells were incubated with 100 µL of DMEM containing 60 µM dichlorodihydrofluorescein diacetate (DCFH-DA) for one hour. Following this, cells were rinsed using PBS and exposed to different sample concentrations (0.250, 0.750 mg mL^−1^) in DMEM for one additional hour. Cells were then washed using PBS and treated with 100 µL of 500 µM 2,2′-azobis(2-amidinopropane) dihydrochloride (AAPH) to induce oxidative stress. Fluorescence from the generated dichlorodihydrofluorescein (DCFH) was measured at an emission wavelength of 538 nm and an excitation of 485 nm, at 2-min intervals over a span of 120 min at 37 °C. Control wells comprised DMEM and DCFH-DA without AAPH, whereas negative controls only contained cells along with DCFH-DA and AAPH. The *CAA* was calculated as a percentage using the following equation:(2)$$CAA\;\left(\%\right)=\left(1-\frac{\int SA}{\int CA}\right)\times 100$$
where ∫*SA* is the integrated area under the sample fluorescence versus time curve, and ∫*CA* is the integrated area from the control curve.

### 2.9. Statistical Analysis

The response variables were analyzed with general linear model (GLM) factorial ANOVA, and the assumptions of normal error distribution and variance homogeneity were verified. Also, the post hoc tests were applied with Student’s all pairwise multiple comparisons method and orthogonal contrasts with an α of 0.05. The results in tables and graphs are presented as means and standard errors. Different letters above the mean bars indicate the differences, and the *F* and *p* values of the model interaction are presented for each statistical model. All data were analyzed using R studio 1.4.1103 (R Core Team).

## 3. Results 

### 3.1. Characterization of Amaranth and Cañihua Protein Concentrates

The composition of the amaranth protein concentrate was as follows: 4.61% moisture, 2.10% lipids, 3.64% ash, and 10.58% fiber. On the other hand, the protein concentrate of cañihua showed 3.25% moisture, 1.91% lipids, 3.83% ash, and 11.19% fiber, and the protein content of the freeze-dried amaranth and cañihua concentrates was 72.81% and 75.22%, respectively. 

Previous studies on protein extracts of amaranth, using alkaline solubilization followed by acid precipitation, indicated a protein content of 52.56% [33]. However, using isoelectric precipitation as an extraction method, Sabbione et al. [34] reported a protein concentration of 73.80% in the amaranth isolate, which aligns closely with our study. As for cañihua, some studies have reported slightly higher protein concentrations. For instance, the method of alkaline solubilization followed by acid precipitation yielded a total protein content of 78.91% [24]. Similarly, Betalleluz-Pallardel et al. [35] reported a concentration of 79.80% under the same extraction conditions. These variations could emerge due to inherent differences in protein content across cultivars. Moreover, factors like agricultural practices, soil conditions, and extraction methodologies may contribute to these discrepancies.

### 3.2. Effect of In Vitro Digestion on Amaranth and Cañihua Protein Concentrates

#### Digestibility of Amaranth and Cañihua Proteins

The digestibility of proteins, particularly those derived from plant sources, is critical in determining their nutritional value and subsequent bioavailability. In this study, when proteins from *A. cruentus* were subjected to in vitro gastrointestinal digestion, there was a pronounced progressive hydrolysis across the various stages and time intervals of digestion (Figure 1). During the initial phase, after 30 min of gastric digestion, the observed digestibility was 60.31%. This value increased slightly to 66.25% by the end of the 120-min gastric digestion period. Notably, after 240 min of in vitro gastrointestinal digestion, the digestibility of the amaranth proteins rose sharply to 79.19%. This enhanced digestibility suggests a combined enzymatic activity of pepsin and pancreatin, pivotal in further degrading high-molecular-weight proteins into shorter peptide chains and free amino acids more readily absorbable by the human digestive system [36]. 

Also, the digestibility of cañihua proteins increased from 41.76% after 30 min of gastric digestion to 71.22% during 240 min of gastrointestinal digestion, indicating that proteins were hydrolyzed into smaller peptides and free amino acids (Figure 1). Several studies have delved into the in vitro digestibility of amaranth proteins. Specifically, *A. cruentus* and *A. caudatus* proteins demonstrated digestibilities of 82% [33] and 82.6% [37], respectively. Furthermore, when germinated, *A. caudatus* exhibited digestibility between 75.0% and 77.2% [38]. These findings underscore the pivotal role played by digestive enzymes in the hydrolysis process. Specifically, pepsin is instrumental in cleaving peptide bonds, especially those near amino acid residues like phenylalanine, tyrosine, tryptophan, and leucine [39]. The hydrolysis of proteins continues through the action of pancreatin, recognized for its extensive range of protein cleavage sites, which facilitates a more comprehensive hydrolysis of proteins [40]. This has significant implications for diets that include amaranth and cañihua, suggesting they can be valuable sources of easily digestible and bioavailable proteins. The results of electrophoresis (SDS-PAGE) for amaranth and cañihua proteins are presented in Figure 2. The amaranth protein concentrate exhibited bands ranging from 20 to 100 kDa (Figure 2A, lane 2), with prominent bands observed at 50–65 kDa and 20–25 kDa. Previous studies on amaranth proteins from the *A. cruentus* and *A. caudatus* species also reported several polypeptides between 14 and 94 kDa, with bands showing higher intensity at 78, 55, 43, 36, 27, and 17 kDa [41]. Amaranth proteins consist of approximately 40% albumin, and the multiple bands observed between 25 and 50 kDa likely correspond to albumin fractions. These fractions have been identified at 34 kDa [42], 30 kDa [43], and 27 kDa [44]. Globulins comprise about 20% of amaranth proteins and are classified as globulins 7S (comamarantine) and 11S (amaranthine). 

The *A. cruentus* species of amaranth has reported globulins at 20–43 kDa [45]. The acid subunits of globulin 11S were identified at 29.3 and 32 kDa, while the basic subunits were found at 20.8 kDa [14]. Additionally, a globulin-type protein with a molecular weight of 64.5 kDa was identified in amaranth proteins of the *A. cruentus* species [43]. The glutelin and prolamine fractions constitute approximately 25–30% and 2–3% of the amaranth proteins, respectively, and their molecular weights have been identified as 67–43 kDa and 110–200 kDa, respectively [45,46]. Gastrointestinal digestion resulted in a decrease in band intensity with longer digestion times since, after 120 min of digestion with pepsin (Figure 2A, lane 4), fainter bands were observed between 48 and 75 kDa. However, no intense bands were observed after intestinal digestion with pancreatin for 60 and 120 min (Figure 2A, lanes 5 and 6), indicating the hydrolysis of proteins into peptides of lower weight and amino acids.

Figure 2B presents the protein characterization of the cañihua protein concentrate and its digestion in different phases and times. In the cañihua protein concentrate, bands from molecular weight between 20 and 100 kDa were visualized, being more intense than the bands between 20 and 50 kDa (Figure 2B, lane 2). Previous studies on cañihua proteins, particularly from Ramis and Cupi-Sayhua species, reported several polypeptide bands between 5 and 250 kDa, and similar bands have been reported corresponding to the albumin fraction, with double bands at 25, 36 and 46 kDa [22]. The 7S globulin fraction was identified with higher intensity bands between 20 and 55 kDa, double bands at 31–35 kDa and 21–27 kDa, and lower intensity between 50 and 92 kDa in the cañihua proteins. Likewise, the 11S globulin fraction was visualized in bands of higher intensity between 4 and 37 kDa and double bands at 31–34 kDa and 21–25 kDa. The bands with lower intensity and molecular weight between 110–200 kDa and 6.5–23 kDa could correspond to prolamins since they have been identified with that molecular weight in previous studies. Likewise, glutelins could correspond to bands with lower intensities and a molecular weight range of 60–230 kDa and 6–37 kDa and double bands at 31–35 kDa and 20–25 kDa [22].

The effect of in vitro gastrointestinal digestion on cañihua proteins shows that proteins withstand the gastric phase for 60 min as the bands remain intact (Figure 2B, lane 3). However, when exposed to the action of the enzyme pepsin for 120 min (lane 4), most of the bands are hydrolyzed. During the intestinal phase (lanes 5 and 6), the sequential action of pepsin and pancreatin resulted in the complete degradation of proteins after 60 min, highlighting the importance of gastrointestinal digestion in releasing peptides and amino acids from cañihua proteins.

### 3.3. Effect of In Vitro Digestion on the Release of Amino Acids from Amaranth and Cañihua Proteins

Table 1 shows the effect of in vitro digestion on releasing amino acids from amaranth proteins. The free amino acid content at the end of gastric digests was 12.302 ± 0.06 mg g^−1^ protein. All essential amino acids were present except for tryptophan. This was not identified due to the analysis conditions causing its degradation through acid hydrolysis. Likewise, during hydrolysis, Asn and Gln tend to undergo deamination and are detected as Asp and Glu, respectively. After gastrointestinal digestion, the free amino acid content significantly increased to 158.344 ± 0.06 mg g^−1^ protein. Among the essential free amino acids, leucine, phenylalanine, and lysine were the most abundant at the end of intestinal digestion of amaranth proteins, with values of 16.131 ± 0.07, 14.738 ± 0.07, and 13.213 ± 0.07 mg g^−1^ protein, respectively. Regarding the non-essential amino acids, tyrosine, arginine, and glutamic acid were predominant in the intestinal digests with values of 16.408 ± 0.07, 15.849 ± 0.07, and 15.084 ± 0.07 mg g^−1^ protein, respectively. Similarly, hydrophobic amino acids constituted the majority in gastric and intestinal digests, with concentrations of 5.606 ± 0.13 and 80.649 ± 0.13 mg g^−1^ protein, respectively. In addition, positively charged amino acids were also predominant in amaranth proteins, with values of 2.676 ± 0.13 and 33.744 ± 0.13 mg g^−1^ protein in the gastric and intestinal phases, respectively, as detailed in Table 1. 

When comparing the content of free amino acids in the gastric and intestinal digests, the latter revealed a significant increase in the concentration of essential amino acids such as valine and methionine, which increased by 61 and 27 times, respectively. On the other hand, non-essential free amino acids like tyrosine and arginine increased 16 and 15 times, respectively, in their concentrations after gastrointestinal digestion. In this context, our results suggest that amaranth proteins are not only highly digestible but also release a balanced profile of both essential and non-essential amino acids during in vitro digestion.

Table 2 shows the effect of in vitro digestion on the content of free amino acids in cañihua proteins. The free amino acid content in the gastric digests was 9.99 ± 0.06 mg g^−1^ protein, and in the gastrointestinal digests, it increased significantly to 149.803 ± 0.06 mg g^−1^ protein. All essential amino acids were found in both gastric and gastrointestinal digests, except tryptophan, which was degraded during analysis. Hydrophobic amino acids were the most abundant in gastric and intestinal digests, followed by positively charged ones. Among the essential free amino acids, phenylalanine, lysine, and leucine were the most abundant at the end of intestinal digestion of amaranth proteins, with values of 15.589 ± 0.04, 13.339 ± 0.04, and 11.085 ± 0.04 mg g^−1^ protein, respectively.

Regarding the non-essential amino acids, arginine, following glutamic and aspartic acids, was predominant in the intestinal digests with values of 16.667 ± 0.04, 16.073 ± 0.04, and 14.726 ± 0.04 mg g^−1^ protein, respectively, in their concentrations after gastrointestinal digestion. In both gastric and intestinal digestion, it was observed that hydrophobic amino acids predominated, with concentrations of 6432 ± 0.06 mg g^−1^ protein and 67,417 ± 0.06 mg g^−1^ protein, respectively. In addition, cañihua proteins were characterized by containing a remarkable amount of positively charged amino acids, presenting concentrations of 1320 ± 0.06 mg g^−1^ protein in the gastric phase and 34,183 ± 0.06 mg g^−1^ protein in the intestinal phase. Upon comparing the gastric and intestinal digests of amaranth proteins, a significant increase in free amino acid content was observed during the intestinal phase. Essential amino acids such as lysine and isoleucine increased 90 and 64 times, respectively. In a similar vein, the non-essential amino acids glycine and arginine increased their concentrations 57 and 27 times, respectively, after simulated gastrointestinal digestion.

All amino acids showed significantly higher values (*p* < 0.05) after in vitro gastrointestinal digestion. Leucine, lysine, arginine, and phenylalanine were the most abundant free amino acids in the digests of amaranth and cañihua. These results are important as plant-based proteins are often considered less favorable than animal-based proteins due to imbalances in amino acid composition and profile [47]. However, both amaranth and cañihua demonstrated an excellent percentage of essential amino acids, and they also contained lysine, which is a limiting amino acid in most cereals [48].

### 3.4. Impact of In Vitro Digestion on the Antioxidant Activity of Amaranth and Cañihua Protein Concentrates

The impact of gastrointestinal digestion on the antioxidant activity of amaranth and cañihua protein concentrates was evaluated by studying the antioxidant activity before and after digestion using various biochemical assays. It is recommended to employ multiple methods to assess the antioxidant activity of a compound in order to understand its potential mechanisms of action under different assay conditions. In this study, the ABTS^•+^, DPPH, and peroxyl (ROO^•^) radical scavenging activities of the amaranth and cañihua concentrates, their digests, and ultrafiltered fractions (<5 kDa and >5 kDa) were analyzed.

#### 3.4.1. Oxygen Radical Absorbance Capacity (ORAC) Assay

The antioxidant activity of amaranth proteins exhibited its highest value in the undigested protein concentrate and remained consistently high throughout the in vitro gastrointestinal digestion process (Table 3). The cañihua proteins demonstrated greater antioxidant capacity during intestinal digestion (Table 3), indicating that the hydrolysis of cañihua proteins is responsible for generating smaller peptides and releasing amino acids with antioxidants as reported by Gallego et al. [49].

Previous studies on *A. cruentus* proteins have shown that in vitro digestion improves the antioxidant activity measured by ORAC with values of 0.129 µmol TE mg^−1^ and 0.213 µmol TE mg^−1^ in aqueous and methanol extracts, respectively [45]. However, the values obtained in our study were higher than those reported previously. Similarly, studies on *A. caudatus* proteins have reported higher antioxidant activity values by ORAC after 60 min (4.28 µmol TE mg^−1^ protein) and 120 min (3.03 µmol TE mg^−1^ protein) of intestinal digestion [50]. In the case of *A. hypochondriacus* proteins, the highest antioxidant capacity by ORAC was shown after hydrolysis using pepsin for 120 min (217.333 µmol TE mg^−1^ soluble protein) [51], while in vitro intestinal digestion of proteins presented 155.2 µmol TE mg^−1^ soluble protein. It has been proposed that the peptides generated during enzymatic hydrolysis inhibit the initiation or propagation of radical reactions by donating hydrogen atoms. This mechanism may explain the observed increase in antioxidant activity by ORAC during the in vitro digestion of *A. mantagazzianus* proteins [14,52]. The variation in antioxidant capacity could be attributed to different amaranth species and digestion conditions [51].

Previous studies on quinoa proteins have reported antioxidant activity values by ORAC of 0.42 µmol TE mg^−1^, which increased to 1.03 µmol TE mg^−1^ after 60 min of pepsin incubation during in vitro gastrointestinal digestion. The highest antioxidant activity was observed after 120 min of intestinal digestion (2.39 µmol TE mg^−1^), indicating that the peptides released during the initial intestinal phase were potent antioxidant agents that maintained their activity even after hydrolysis with pancreatin until the end of digestion [47]. Similarly, another study reported the highest antioxidant activity by ORAC in papain-hydrolyzed quinoa proteins (0.501 µmol TE mg^−1^) [53]. The differences in results can be attributed not only to the inherent nature of the native protein but also to the generation and release of bioactive sequences by different enzymes employed in the hydrolysis process [54,55].

#### 3.4.2. ABTS Radical Scavenging Activity

The ABTS radical removal properties of amaranth and cañihua proteins were evaluated at different stages of in vitro digestion. In vitro digestion of amaranth proteins significantly increased their antioxidant activity compared to undigested proteins, with the intestinal digestion phase being the most effective (Table 3). Similarly, in vitro digestion of cañihua proteins markedly enhanced their antioxidant activity, suggesting the release of bioactive peptide sequences capable of inhibiting the ABTS radical (Table 3). In this sense, it has been reported that the hydrolysis of *A. caudatus* proteins using the Alcalase-Neutrase combination for 240 min exhibited a high antioxidant activity (1.67 μmol TE mg^−1^ protein) [25]. Major antioxidant activity values have also been recorded in protein hydrolysates of *A. hypochondriacus* with 90 min of incubation with the enzyme flavourzyme (0.0042 mg TE mg^−1^) [56]. On the other hand, when assessing the antioxidant properties of cañihua proteins via the ABTS method, it was observed that activity augmented in tandem with the duration of enzymatic hydrolysis. A synergistic utilization of flavourzyme and alcalase enzymes for 240 min culminated in the most potent antioxidant activity [24].

Increased antioxidant activity has been associated with the release of peptides and amino acids which will depend on the chemical structure of the protein. In addition, the sequential use of enzymes promotes the production of peptides with antioxidant activity [25]. The presence of aromatic amino acids such as tryptophan and tyrosine also contributes to their antioxidant activity by donating electrons to free radicals [57,58]. In addition, the generation of peptides with antioxidant activity depends not only on the nature of the native protein but also on the enzymes used for its hydrolysis [59]. The results obtained in this study show that gastrointestinal digestion favors the ABTS scavenging activity. However, the antioxidant activity of the samples can be influenced by several factors, among which are the concentration of proteins, the type of proteins, the extraction method, the conditions under which hydrolysates were obtained, and gastrointestinal digestion in vitro, among others [60].

#### 3.4.3. DPPH Radical Scavenging Activity

The ability of amaranth and cañihua proteins to scavenge DPPH radicals was evaluated. Notably, the peak antioxidant activity was observed during the intestinal phase, while the nadir was evident in the undigested proteins of both food sources (Table 3). Consistent with our findings, previous studies have reported increased antioxidant activity in *A. cruentus* proteins after in vitro digestion. Specifically, a study by [45] registered values of 0.0197 µmol TE mg^−1^ and 0.01217 µmol TE mg^−1^ in aqueous and methanol extracts, respectively. Similarly, the highest antioxidant activity has been observed in hydrolyzed A. *hypochondriacus* proteins using alcalase and flavourzyme (0.0038 µmol TE mg^−1^) [56], while the activity was lower in undigested proteins (0.00076 µmol TE mg^−1^).

Specific amino acid sequences, such as LVRW with RW amino acids at the C-terminal end, have been associated with antioxidant activity, and sequences ranging from 5 to 16 amino acids have demonstrated antiradical activity [61,62,63,64,65]. The amount, structure, and size of peptides generated during enzymatic hydrolysis can significantly influence antioxidant activity [66]. Smaller peptides have been found to have a higher electron-donating capacity compared to larger peptides [67]. During digestion, the bioactive sites of the peptides are exposed, allowing them to interact with the DPPH radical more effectively [68]. Aromatic residues play a crucial role in neutralizing the electron-donating radical, and amino acids such as Cys, Met, and His have demonstrated the ability to eliminate free radicals, reduce oxidation, and bind metals [49], while histidine is known for its electron-donating properties [69].

#### 3.4.4. Cellular Antioxidant Activity (CAA)

The CAA allows measuring intracellular ROS using 2′,7′-dichlorofluorescin diacetate (DCFH-DA). DCFH-DA is a derivative of 2′,7′-dichlorofluorescin (DCFH) that can readily traverse cell membranes because of its non-polar and non-ionic characteristics. Once inside the cell, endogenous cellular esterases act on DCFH-DA, converting it back to DCFH. This form is more prone to oxidation and remains inside the cell due to its polarity. When ROS are present, they oxidize DCFH, transforming it into its fluorescent form, 2′,7′-dichlorofluorescein (DCF) [70]. In antioxidant analysis, especially in nutritional studies that delve into bioactive compounds and nutraceuticals, the human hepatocellular carcinoma (HepG2) cell line is a predominant choice [71]. Another prominent choice in antioxidant-related studies is the CaCo-2 cell line, recognized for its affinity with active membrane transport in the intestinal cell wall [70,72].

To evaluate the cytotoxicity and antioxidant activity of amaranth and cañihua proteins and their gastric and intestinal digests, HepG2 and CaCo-2 cell lines were used. Cell viability exceeded 90% for concentrations ranging from 0.125 to 1 mg mL^−1^ in both cell lines, indicating the non-toxic nature of the samples at these specified doses. Subsequently, cellular antioxidant activity was evaluated in HepG2 and CaCo-2 cell lines at concentrations of 0.250 and 0.750 mg mL^−1^. Therefore, for an antioxidant to prevent DCFH to DCF oxidation, it must permeate the cell and compete intracellularly with AAPH free radicals. Our results show that amaranth and cañihua proteins, after gastrointestinal digestion, confer homogeneous and gradual cellular antioxidant protection that increases significantly with concentration (*p* < 0.05). In both protein concentrates, no statistical differences were observed at the concentration of 0.250 mg mL^−1^. However, the highest CAA was observed at the higher concentration of 0.750 mg mL^−1^.

In amaranth proteins, the highest CAA in liver cells (HepG2) was observed in the intestinal fractions < 5 kDa (81.05%), intestinal fractions > 5 kDa (81.02%), and gastric fractions < 5 kDa (79.81%) (Figure 3A). Similarly, the highest CAA in intestinal cells (Caco-2) was observed in the intestinal fractions < 5 kDa (84.68%), intestinal fractions > 5 kDa (84.60%), and gastric fractions < 5 kDa (82.50%) (Figure 3B). In the case of cañihua proteins, the highest CAA in liver cells (HepG2) was observed in the intestinal fractions < 5 kDa (82.57%), intestinal fractions > 5 kDa (81.06%), and gastric fractions > 5 kDa (80.44%) (Figure 3A). Similarly, in intestinal cells (Caco-2), all fractions had no significantly different high CAA ranging from 82.31% in GD < 5 kDa and 85.55% in ID < 5 kDa for a concentration of 0.750 mg mL^−1^ (Figure 3B). The protein concentrates without digestion (D0) showed the lowest CAA in both cell lines (Figure 3A,B). 

Previous studies have reported the cytoprotective effect of oat protein hydrolysates in liver cells (HepG2) by reducing oxidative enzymes and intracellular ROS induced by AAPH. This oxidation inducer generates peroxyl radicals, causing cellular damage [73]. The authors associated this effect with the ability of bioactive peptides to reduce hydroperoxides and convert oxidized oxygen to less reactive hydrogen peroxide, which can be further reduced to water by endogenous enzymes such as catalase. Similarly, the cellular antioxidant activity of peptide fractions (<1 kDa) from orchid proteins (*Dendrobium aphyllum*) has been demonstrated in liver cells (HepG2), indicating that low-molecular-weight peptides can penetrate the cell membrane and reduce intracellular levels of ROS by inducing the intracellular system of antioxidant enzymes [74]. Corn peptides have also shown cytoprotective effects in HepG2 cells by eliminating free radicals, stabilizing the cell membrane, and enhancing the activity of endogenous antioxidant enzymes [75].

In intestinal cells (Caco-2), *A. mantegazzianus* proteins and their fractions have been reported to reduce oxidative stress by intracellular ROS and prevent oxidative damage through various mechanisms, such as the use of low-weight compounds capable of eliminating ROS by donating electrons or hydrogens and restoring signaling pathways that induce the activity of antioxidant enzymes [76]. Peptides with residues between 5 and 16 amino acids have been shown to cross the intestinal barrier and interact with free radicals, indicating their potential role in reducing intracellular ROS formation [54]. The differences in antioxidant activity observed between cell lines may be attributed to variations in the cell membranes’ composition, surface, and fluidity [40]. The cytoprotective effects of peptides can be attributed to their ability to reduce reactive oxygen species and enhance the activity of endogenous antioxidant enzymes [11]. The concentration of protein, peptides, and free amino acids is also related to their antioxidant potency [77].

Finally, although cell lines provide valuable information and allow the controlled study of specific mechanisms, they do not fully reflect the complex functions and interactions that occur in vivo. Therefore, further studies in vitro and in vivo are needed to better understand the impact and mechanisms of action of these bioactive compounds in a broader biological context.

## 4. Conclusions

Our study shows that amaranth and cañihua proteins exhibit antioxidant activity, enhanced by gastrointestinal digestion. This protective effect shows a dose–response relationship, as cellular antioxidant activity becomes more substantial with increasing protein concentration. In particular, antioxidant activity reaches a maximum of 85% in HepG2 and Caco-2 cell lines for intestinal fractions at 0.750 mg mL^−1^. This study highlights digestion’s critical role in releasing bioactive amino acids and peptides with antioxidant properties. These findings advance our understanding of the health benefits of amaranth and cañihua protein consumption. In addition, the release of amino acids during each stage of digestion provides crucial data for nutritional research focused on pseudocereal protein concentrates. These protein concentrates have significant potential as nutraceutical ingredients that can promote health and combat cellular oxidative stress associated with chronic disease development. Future steps will focus on identifying the specific peptide sequences that contribute to the antioxidant effects observed in this study.

## Figures and Tables

**Figure 1 antioxidants-12-02075-f001:**
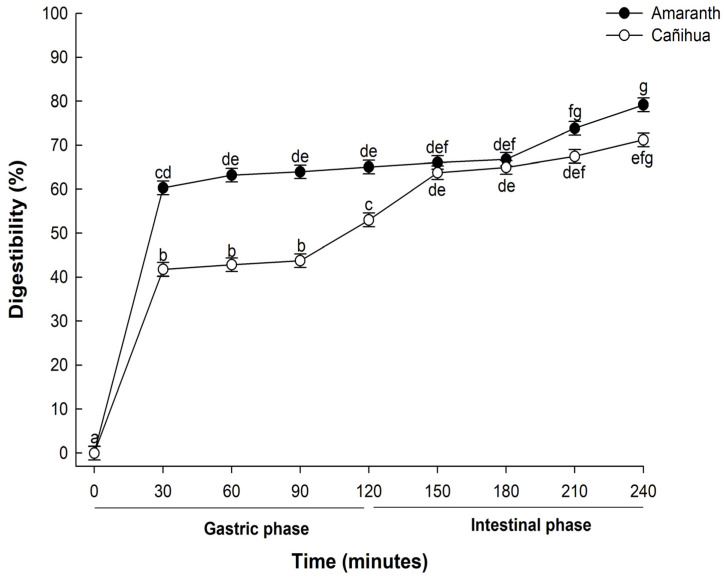
Mean percentage of digestibility (±standard error) of amaranth (*A. cruentus*) and cañihua (*Ch. pallidicaule*) proteins over time obtained with two-way ANOVA (Appendix A). The letters indicate the multiple comparisons between the orthogonal contrast method food matrix and time for α of 0.05.

**Figure 2 antioxidants-12-02075-f002:**
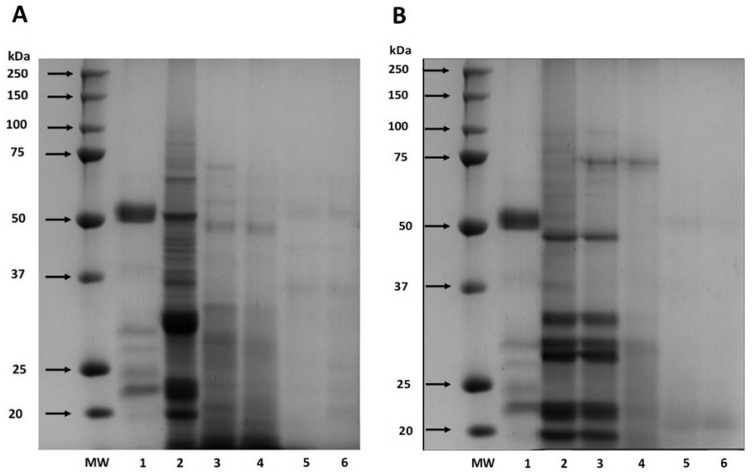
SDS-PAGE protein profiles of amaranth (*A. cruentus*) protein concentrate (**A**) and cañihua (*Ch. pallidicaule*) protein concentrate (**B**) at different times of simulated gastrointestinal digestion under reducing conditions. MW: molecular weight marker where lane 1 shows digestion blank with digestive enzymes, lane 2 shows undigested protein, lane 3 shows gastric digest at 60 min, lane 4 shows gastric digest at 120 min, lane 5 shows gastrointestinal digest at 60 min, and lane 6 shows gastrointestinal digest at 120 min.

**Figure 3 antioxidants-12-02075-f003:**
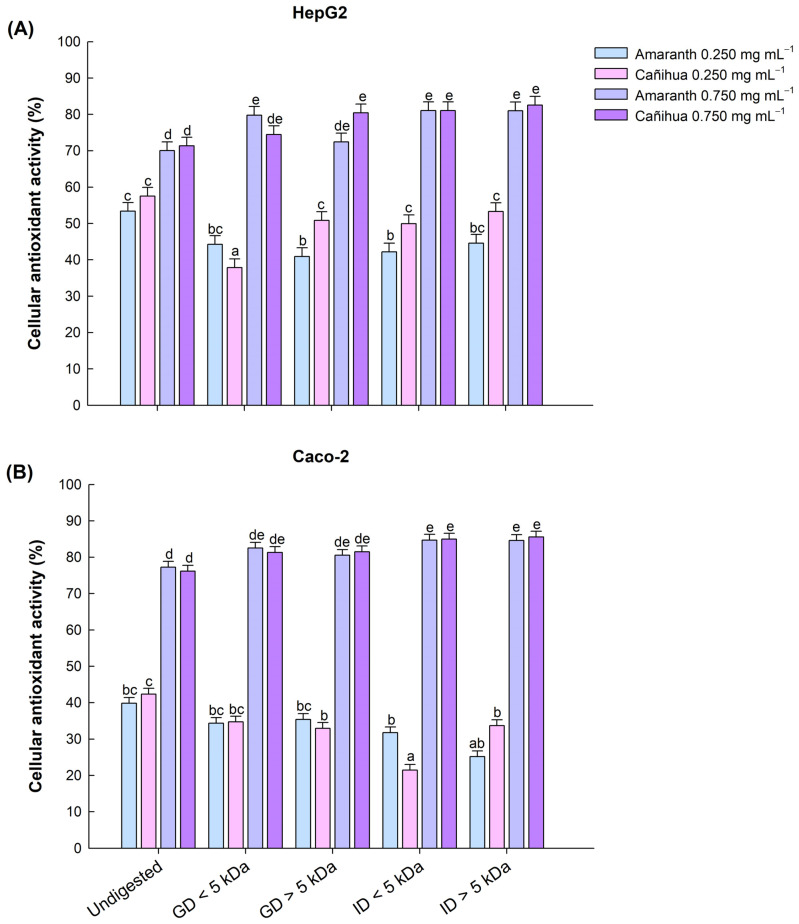
Effect of gastric and intestinal digests of amaranth and cañihua proteins with molecular weight < 5 kDa and >5 kDa on antioxidant activity (%) in human hepatocellular carcinoma cells (HepG2) (**A**) and human colorectal adenocarcinoma cells (Caco-2) (**B**). Comparison between protein concentrate without digestion (Undigested), gastric digestion fractions < 5 kDa (GD < 5 kDa), gastric digestion fractions > 5 kDa (GD > 5 kDa), intestinal digestion fractions < 5 kDa (ID < 5 kDa), and intestinal digestion fractions > 5 kDa (ID > 5 kDa). The mean values (±standard error) obtained from three-way ANOVA (Appendix A) are shown, and the letters on the bars indicate the post hoc for α of 0.05 between the digests at the different protein concentrations.

**Table 1 antioxidants-12-02075-t001:** Profile of free amino acids (mg g^−1^ protein) found at the end of gastric and intestinal digestion phases of *A. cruentus* proteins.

Amino Acid	Gastric Digestion	Intestinal Digestion
Val	0.143 ± 0.07 ^a^	8.710 ± 0.07 ^h^
Met	0.173 ± 0.07 ^a^	4.626 ± 0.07 ^f^
Gly	0.230 ± 0.07 ^ab^	2.970 ± 0.07 ^e^
Cys	0.289 ± 0.07 ^ab^	1.871 ± 0.07 ^d^
Pro	0.454 ± 0.07 ^ab^	2.882 ± 0.07 ^e^
Ala	0.460 ± 0.07 ^ab^	5.969 ± 0.07 ^g^
Thr	0.481 ± 0.07 ^ab^	4.664 ± 0.07 ^f^
His	0.604 ± 0.07 ^b^	4.682 ± 0.07 ^f^
Asp	0.854 ± 0.07 ^bc^	12.058 ± 0.07 ^k^
Phe	0.867 ± 0.07 ^bc^	14.738 ± 0.07 ^l^
Ile	0.903 ± 0.07 ^bc^	9.314 ± 0.07 ^i^
Ser	0.951 ± 0.07 ^bc^	9.175 ± 0.07 ^i^
Arg	1.022 ± 0.07 ^bc^	15.849 ± 0.07 ^m^
Tyr	1.049 ± 0.07 ^c^	16.408 ± 0.07 ^n^
Lys	1.050 ± 0.07 ^c^	13.213 ± 0.07 ^j^
Leu	1.268 ± 0.07 ^cd^	16.131 ± 0.07 ^mn^
Glu	1.504 ± 0.07 ^d^	15.084 ± 0.07 ^l^
EAA	5.489 ± 0.13	76.078 ± 0.13
Aromatic	1.916 ± 0.13	30.869 ± 0.13
Branched chain	2.314 ± 0.13	34.155 ± 0.13
Hydrophobic	5.606 ± 0.13	80.649 ± 0.13
Positively charged	2.676 ± 0.13	33.744 ± 0.13
Negatively charged	2.371 ± 0.13	27.142 ± 0.13

Essential amino acids (EAA) (Val + Met + Thr + His + Phe +Ile + Lys + Leu), Aromatic (Phe + Tyr), Branched chain (Leu + Ile + Val), Hydrophobic (Ala + Val + Ile + Leu + Tyr + Phe + Pro + Met + Cys), Positively charged (Arg + His + Lys), and Negatively charged (Asp + Glu). Mean values (±standard error) for each amino acid are expressed in mg g^−1^ protein after both gastric and intestinal digestion phases. Different letters indicate the multiple comparisons by using the method of orthogonal contrasts (α = 0.05) obtained from the two-way ANOVA (Appendix A).

**Table 2 antioxidants-12-02075-t002:** Profile of free amino acids (mg g^−1^ protein) found at the end of gastric and intestinal digestion phases of *Ch. pallidicaule* proteins.

Amino Acid	Gastric Digestion	Intestinal Digestion
Gly	0.067 ± 0.04 ^a^	3.818 ± 0.04 ^g^
Ile	0.088 ± 0.04 ^a^	5.662 ± 0.04 ^j^
Lys	0.148 ± 0.04 ^a^	13.339 ± 0.04 ^o^
Cys	0.154 ± 0.04 ^a^	1.717 ± 0.04 ^f^
Ala	0.194 ± 0.04 ^a^	5.822 ± 0.04 ^k^
Met	0.291 ± 0.04 ^ba^	3.984 ± 0.04 ^g^
Pro	0.411 ± 0.04 ^b^	4.103 ± 0.04 ^hg^
Thr	0.452 ± 0.04 ^b^	4.865 ± 0.04 ^i^
Val	0.462 ± 0.04 ^b^	6.250 ± 0.04 ^l^
His	0.552 ± 0.04 ^b^	4.177 ± 0.04 ^hg^
Arg	0.620 ± 0.04 ^b^	16.667 ± 0.04 ^s^
Asp	0.648 ± 0.04 ^b^	14.726 ± 0.04 ^p^
Ser	0.756± 0.04 ^b^	10.721 ± 0.04 ^m^
Tyr	1.132 ± 0.04 ^c^	13.205 ± 0.04 ^o^
Phe	1.235± 0.04 ^d^	15.589 ± 0.04 ^q^
Glu	1.315± 0.04 ^d^	16.073 ± 0.04 ^r^
Leu	1.465 ± 0.04 ^ed^	11.085 ± 0.04 ^n^
EAA	4.693 ± 0.06	64.951± 0.06
Aromatic	2.367 ± 0.06	28.794 ± 0.06
Branched chain	3.015 ± 0.06	22.997 ± 0.06
Hydrophobic	6.432 ± 0.06	67.417 ± 0.06
Positively charged	1.320 ± 0.06	34.183 ± 0.06
Negatively charged	1.963 ± 0.06	30.799 ± 0.06

Essential amino acids (EAA) (Val + Met + Thr + His + Phe +Ile + Lys + Leu), Aromatic (Phe + Tyr), Branched chain (Leu + Ile + Val), Hydrophobic (Ala+ Val + Ile + Leu + Tyr + Phe + Pro + Met + Cys), Positively charged (Arg + His + Lys), and Negatively charged (Asp + Glu). Mean values (±standard error) for each amino acid are expressed in mg g^−1^ protein after both gastric and intestinal digestion phases. Different letters indicate the multiple comparisons by using the method of orthogonal contrasts (α = 0.05) obtained from the two-way ANOVA (Appendix A).

**Table 3 antioxidants-12-02075-t003:** Mean values (±standard error) of antioxidant activity at different stages of in vitro gastrointestinal digestion of amaranth and cañihua proteins.

Sample	ORAC Value (µmol TE mg^−1^)		ABTS Value (µmol TE mg^−1^)		DPPH Value (µmol TE mg^−1^)	
	Amaranth	Cañihua	Amaranth	Cañihua	Amaranth	Cañihua
Protein concentrate	1.78 ± 0.01 ^d^	1.42 ± 0.01 ^b^	29.80 ± 1.28 ^a^	29.82 ± 1.28 ^a^	3.48 ± 0.19 ^a^	5.75 ± 0.36 ^b^
Gastric digestion	1.57 ± 0.01 ^c^	1.28 ± 0.01 ^a^	119.97 ± 1.28 ^bc^	119.05 ± 1.28 ^bc^	5.58 ± 0.19 ^b^	6.88 ± 0.36 ^b^
Intestinal digestion	1.62 ± 0.01 ^c^	1.45 ± 0.01 ^b^	121.97 ± 1.28 ^c^	114.89 ± 1.28 ^b^	6.38 ± 0.19 ^b^	12.18 ± 0.36 ^c^

The letters show multiple comparisons by using the method of orthogonal contrast (α = 0.05) obtained after the two-way ANOVA (Appendix A), and the different antioxidant assays were analyzed separately. TE: Trolox equivalents.

## Data Availability

The data are contained within the article and Appendix A.

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
