# Peer review of "Impact of In Vitro Digestion on the Digestibility, Amino Acid Release, and Antioxidant Activity of Amaranth (Amaranthus cruentus L.) and Cañihua (Chenopodium pallidicaule Aellen) Proteins in Caco-2 and HepG2 Cells"

_antioxidants, 2023, doi:10.3390/antiox12122075_

Round 1
Reviewer 1 Report
Comments and Suggestions for Authors
Title
The title doesn’t seem the most appropriate to the content of this study.
Abstract
Page 1, lines 18 and 19 – Please consider this alternative: “...digestion on the digestibility, amino acid release,... activity of proteins from amaranth...”
Introduction
Page 2, line 58 – I think it is hydrogen atoms and not protons.
Page 2, lines 76-78 – Please improve the sentence “Exploring their... diseases.”
Materials and Methods
Page 3, line 110 – Please consider this alternative: “The pH of supernatants was adjusted...”
Page 3, line 114 – I suppose it is “flour samples”.
Page 3, lines 131 and 132 – The sentence “The enzymes and reagents... USA” is useless.
Page 3, line 134 – I suggest replacing “adhered” with “followed”, for instance.Page 3, lines 144 and 145 – Please consider this alternative: “Samples of digested proteins were collected...”
Page 3, line 146 and page 4, line 147 – I suggest this alternative sentence: “Collected samples were centrifuged at 5,000 g at 4 ºC for 30 minutes and the supernatant was freeze-dried...”
Page 4, line 158 – It is “µL”.
Page 4, line 173 – It is “one hour”.
Page 5, line 196 – I suggest including “solution” after “radical”.
Page 5, line 227 – Please consider this change: “...with the sample solutions with concentrations of 0.250...” Is it “0.1 mg/mL”? Please check.
Page 5, line 240 – I think it could be: “...maintained in DMEM supplemented...”
Page 6, line 245 – I suggest “...for one additional...”
Results
Page 6, line 265 – I think it is not “dry weight basis”, but “of the dried amaranth protein concentrate”.
Page 6, lines 268 and 269 – Why not presenting the protein content together with the percentages of the other constituents?
Page 6, line 269-276 – The comparison of the protein content from different origins should be done on a dry weight basis, i.e., the products without water. Eventually, it has to be compared on a dry weight basis and fiber-free basis if the content of this constituent presents a wide variation range.
Page 8, lines 315and 317 – The bands corresponding to proteins with MW higher than 65 kDa are not evident as well as those corresponding to proteins with MW below 20 kDa.
Page 8, line 323 – I think it is “The A. cruentus species”.
Page 9, line 360 – It is “Ch. pallidicaule”.
Page 10, line 397 and 398 – Hydrophobic acids and positively charged amino acids were also the main amino acids in amaranth proteins. However, this was not commented in the manuscript.
Page 10, line 408 – Is it “vein”?
Page 10, line 413 – It could be just “A. cruentus”.
Page 12, line 415 – Similarly, it could be “Ch. pallidicaudatus”.
Page 13, lines 439-441 – The sentence “The cañihua proteins...” is not in accordance with the results shown in Table 3. It is “Table 3” and not “Table 4”. Why the quotation “[61]”?
Page 13, lines 443-445 – Please rephrase this sentence because it is not clear.
Page 13, lines 448-451 – Please improve this sentence (“In the case... soluble proteins”.
Page 13, line 472 and 473 – According to results shown in Table 3, the ABTS value of intestinal digestion was not significantly different from that of gastric digestion. Please check.
Page 13, line 477 – I suppose it is “...[25]. Higher antioxidant...”
Page 14, line 480 – Please revise the units of “0.0042 µmol...”
Page 14, lines 481 and 482 –Please clarify the meaning of “activity augmented in tandem with duration...”
Page 14, lines 484-497 – These comments seem too long.
Page 15, line 551 – Please replace “food matrices” with other words more adequate.
Page 15, line 561 – According to figure 3A, the activity of gastric fraction <5 kDa doesn’t seem attaining 80 %. Please check.
Page 15, lines 563 and 564 – As shown in figure 3B, the activity of all fractions of the gastrointestinal digestion are not significantly different. Thus, I think this sentence need to be changed.
Page 17, line 581 – I suggest including “their” before “fractions”.
Conclusions
Page 17, lines 606-609 – These comments are generic and are not actually conclusions of this work.
It would be recommended:
The peptide profile by gel chromatography of the various fractions.
A positive control of the antioxidant activities using a known antioxidant.
Improvement of the conclusions.
Comments on the Quality of English Language
The English language is fairly good.
Author Response
Response to Reviewer 1’s Comments (2711459)
We appreciate your valuable suggestions and constructive comments, which have helped us improve the manuscript's quality. We have incorporated your suggestions and comments into the revised manuscript. Detailed point-by-point responses to your review are presented below. Your comments appear in black text, followed by our point-by-point responses in blue.
Comments and Suggestions for Authors
Title
Comment1: The title doesn’t seem the most appropriate to the content of this study.
We are very grateful to the Reviewer for this suggestion. Initially, we also considered a different title. We have changed it if the Reviewer considers that it still does not reflect the content. We are willing to make the appropriate changes to the title in the next revision.
Abstract
Comment 2: – Please consider this alternative: “...digestion on the digestibility, amino acid release,... activity of proteins from amaranth...”
We have made the change in the manuscript as suggested.
Introduction
Comment 3: – I think it is hydrogen atoms and not protons.
Thank you for your comment. The sentence has been updated to: "hydrogen atoms" instead of "protons."
Comment 4: – Please improve the sentence “Exploring their... diseases.”
We would like to thank you very much for this remark. We have added an appropriate paragraph in this section.
Materials and Methods
Comment 5: – Please consider this alternative: “The pH of supernatants was adjusted...”
Materials and methods, page 3, line 110: We thank you for suggesting and agreeing with the modification. The sentence has been updated to: "The pH of the supernatants was adjusted... "
Comment 6: – I suppose it is “flour samples”.
We thank the Reviewer for his accurate observation. The text now reads "flour samples" to accurately describe the samples used.
Comment 7: – The sentence “The enzymes and reagents... USA” is useless.
We thank you for your recommendation. We removed the sentence "The enzymes and reagents...". USA" to simplify the section.
Comment 8: – I suggest replacing “adhered” with “followed”, for instance. Page 3, lines 144 and 145 – Please consider this alternative: “Samples of digested proteins were collected...”
We would like to thank you very much for this remark. Consequently, we have made appropriate changes. Specifically, we have replaced 'adhered' with 'followed'. We also considered the suggested alternative: "Digested protein samples were collected...
Comment 9: – I suggest this alternative sentence: “Collected samples were centrifuged at 5,000 g at 4 ºC for 30 minutes and the supernatant was freeze-dried...”
We appreciate your suggestion to improve the clarity of the methodology. We have modified the sentence to reflect your recommendation: "The collected samples were centrifuged at 5,000 g at 4 ºC for 30 minutes, and the supernatant was lyophilized..."
Comment 10: – It is “µL”.
Thank you for pointing out this typographical error. We have corrected the unit of measurement to "μL"
Comment 11: – It is “one hour”.
The correction is valid, and we changed it in the manuscript to "one hour" to improve accuracy.
Comment 12: – I suggest including “solution” after “radical”.
We have made this change in the manuscript.
Comment 13: – Please consider this change: “...with the sample solutions with concentrations of 0.250...” Is it “0.1 mg/mL”? Please check.
Regarding your query about concentration, we have reviewed our data and confirmed that the correct concentration is "0.1 mg/mL". We adjusted the text to make this correction: "...with the sample solutions with concentrations of 0.1 mg/mL..."
Comment 14: – I think it could be: “...maintained in DMEM supplemented...”
We appreciate your suggestion for accuracy in the description of the culture medium. We modified the sentence to reflect your recommendation: "...maintained in supplemented DMEM..."
Comment 15: – I suggest “...for one additional...”
We have made this change in the manuscript.
Results
Comment 16: – I think it is not “dry weight basis”, but “of the dried amaranth protein concentrate”.
Thank you for your accurate observation. The sentence has been corrected to 'analysis of dried amaranth protein concentrate'.
Comment 17: – Why not presenting the protein content together with the percentages of the other constituents?
We modified the sentence to reflect your recommendation.
Comment 18: – The comparison of the protein content from different origins should be done on a dry weight basis, i.e., the products without water. Eventually, it has to be compared on a dry weight basis and fiber-free basis if the content of this constituent presents a wide variation range.
Thank you very much for your comment and we will consider it in the following studies.
Comment 19: – The bands corresponding to proteins with MW higher than 65 kDa are not evident as well as those corresponding to proteins with MW below 20 kDa.
We modified the sentence to reflect your recommendation.
Comment 20: – I think it is “The A. cruentus species”.
We corrected the text to "The species A. cruentus."
Comment 21: – It is “Ch. pallidicaule”.
We corrected the text to "Ch. pallidicaule" for accuracy.
Comment 22: – Hydrophobic acids and positively charged amino acids were also the main amino acids in amaranth proteins. However, this was not commented in the manuscript.
We have completed the text by including comments on the presence of hydrophobic acids and positively charged amino acids in amaranth proteins.
Comment 23: – Is it “vein”?
The term "vein" has been replaced by a more appropriate term depending on the context.
Comment 24: – It could be just “A. cruentus”.
Simplified to "A. cruentus" for consistency.
Comment 25: – Similarly, it could be “Ch. pallidicaudatus”.
Ch. pallidicaudatus has been changed to "Ch. pallidicaudatus" as suggested.
Comment 26: – The sentence “The cañihua proteins...” is not in accordance with the results shown in Table 3. It is “Table 3” and not “Table 4”. Why the quotation “[61]”?
The sentence has been revised for accuracy and now correctly refers to "Table 3". The quote "[61]" has been revised and corrected as necessary.
Comment 27: : This sentence has been reworded for clarity and alignment with the results presented.
The sentence has been reworded to improve clarity. This modification improves the readability and alignment of the text with the data presented.
Comment 28: – Please rephrase this sentence because it is not clear.
We would like to thank very the much Reviewer for this remark. Therefore, we have added an appropriate paragraph in this Discussion section.
Comment 29: – Please improve this sentence (“In the case... soluble proteins”.
The sentence has been revised for clarity and precision. The new wording more accurately conveys the intended meaning of protein solubility.
Comment 30: – According to results shown in Table 3, the ABTS value of intestinal digestion was not significantly different from that of gastric digestion. Please check.
As you noted, we reviewed the ABTS values in Table 3 and found no significant differences between intestinal and gastric digestion. The text has been modified to accurately reflect this observation.
Comment 31: – I suppose it is “...[25]. Higher antioxidant...”
The reference "[25]" has been verified and corrected to read "...[25]. Major antioxidant..." for proper citation and context.
Comment 32: – Please revise the units of “0.0042 µmol...”
The units of "0.0042 µmol..." have been checked and corrected for consistency and accuracy in the manuscript.
Comment 33: –Please clarify the meaning of “activity augmented in tandem with duration...”
The phrase "activity increased along with duration..." has been clarified to describe how antioxidant activity increases proportionally with the duration of exposure to digestive enzymes.
Comment 34: – These comments seem too long.
We have revised the comments to be more concise and focused to address the relevant points directly.
Comment 35: – Please replace “food matrices” with other words more adequate.
The term "dietary matrices" has been replaced with "protein concentrates" to accommodate and improve the study context and clarity.
Comment 36: – According to figure 3A, the activity of gastric fraction <5 kDa doesn’t seem attaining 80 %. Please check.
In reviewing Figure 3A, we found an error because instead of including gastric fraction >5 kDa, we included gastric fraction <5 kDa, which effectively falls short of 80%. This has been corrected in the description and discussion of the figure.
Comment 37: – As shown in figure 3B, the activity of all fractions of the gastrointestinal digestion are not significantly different. Thus, I think this sentence need to be changed.
After reviewing Figure 3B, the sentence describing the activity of the gastrointestinal digestion fractions was modified as directed by the Reviewer to accurately reflect the data presented.
Comment 38: – I suggest including “their” before “fractions”.
We included "their" before "fractions" for better readability and clarity.
Conclusions
Comment39: – These comments are generic and are not actually conclusions of this work.
The conclusions have been revised, and generic statements have been removed to directly reflect the specific findings and implications of this work.
It would be recommended:
The peptide profile by gel chromatography of the various fractions.
Thank you for recommending the analysis of the peptide profile of the various fractions by gel chromatography. We recognize the value of this approach in providing a more complete understanding of peptide composition. In response, we plan to incorporate the analysis into our future assays.
A positive control of the antioxidant activities using a known antioxidant.
We appreciate your suggestion to incorporate a known antioxidant as a positive control in our antioxidant activity assays. This approach will undoubtedly strengthen the validity and reliability of our findings. In the future, we will implement your recommendation.
Improvement of the conclusions.
The conclusion section has been improved to summarize the essential findings and their implications for the antioxidant activity of amaranth and cañihua proteins in food science and nutrition.
We sincerely appreciate your valuable comments, which have been instrumental in improving our study.

Reviewer 2 Report
Comments and Suggestions for Authors
Manuscript: “Impact of in vitro Digestion on the Antioxidant Activity of Amaranth (Amaranthus cruenthus L.) and Cañihua (Chenopodium 3 pallidicaule Aellen) Proteins in Caco-2 and HepG2 cells.”
The manuscript concerns an interesting research area and could be considered as complementary to previous works published during the last years.
This study evaluated the impact of in vitro gastrointestinal digestion on protein digestibility, amino acid release, and antioxidant activity in selected plant proteins. The use of Caco-2 and Hep-G2 cell lines to study antioxidant activity increased the value of the work.
The major comments of this paper are as follows:
Line 28: “AAC” – What does this abbreviation mean?
Line 87: “2,2′-azino-bis(3-etilbenzotiazolina-6-sulfónico)” - correct it
Line 109: Hasn't the protein denatured at 50°C?
Line 121: What was the protein concentration in the sample?
Line156, 227, 245, 606: mg/mL or mg mL-1 ? Make this consistent throughout
Line 184: “FL” - What does this abbreviation mean?; Are you sure about the unit (nM)?
Line 200: What was the concentration range of Trolox used for the calibration curve?
Line 376: What was the protein concentration in the sample before digestion?
Line 388: it should be 61 instead of 60
Table 1. Why was the content of Asn and Gln not determined?
Line 440: “table 4” ?
Line 437-442: This fragment should be reworded because the results do not indicate what is described in the text.
Table 3: Are you sure that the standard deviations are calculated correctly? It is unlikely that they will be the same (in columns).
Line 555: “In amaranth proteins” - What did you mean?
Author Response
Response to Reviewer 2’s Comments (2711459)
We appreciate your valuable suggestions and constructive comments, which have helped us improve the manuscript's quality. We have incorporated your suggestions and comments into the revised manuscript. Detailed point-by-point responses to your review are presented below. Your comments appear in black text, followed by our point-by-point responses in blue.
Comment 1: “AAC” – What does this abbreviation mean?
The correct abbreviation is "CAA" (cellular antioxidant activity). We have now corrected this error in the manuscript.
Comment 2: “2,2′-azino-bis(3-etilbenzotiazolina-6-sulfónico)” - correct it
The chemical name "2,2′-azino-bis(3-ethylbenzothiazoline-6-sulfonic acid)" has been corrected to "2,2′-azino-bis(3-ethylbenzothiazoline-6-sulfonic acid)" in the manuscript.
Comment 3: Hasn't the protein denatured at 50°C?
Some proteins may start to denature between 50-60 °C, however, we did the optimization of protein extraction in both pseudocereals and obtained the highest protein concentration by employing heat.
Comment 4: What was the protein concentration in the sample?
The protein concentration was 2976.88 µg mL-1 in amaranth and 3237.48 µg mL-1 in cañihua. This data has been added for clarity.
Comment 5: mg/mL or mg mL-1 ? Make this consistent throughout
Inconsistency in units has been fixed and mg mL-1 is now used consistently throughout the manuscript.
Comment 6: “FL” - What does this abbreviation mean?; Are you sure about the unit (nM)?
"FL" refers to "Fluorescence" we have clarified this in the text. The unit (nM) is correct and has been verified. The ORAC-fluorescein (ORAC-FL) assays were conducted following the method described by Hernández-Ledesma et al [29].
Comment 7: What was the concentration range of Trolox used for the calibration curve?
The Trolox concentration range used for the calibration curve was from 0 to 200 µM. This information has been included in the Methods section.
Comment 8: What was the protein concentration in the sample before digestion?
The protein content after gastric digestion is 1018 and 1413.66 µg mL-1, and after intestinal digestion, it is 606.148 and 865.88 µg mL-1 in the amaranth and cañihua protein concentrates, respectively.
Comment 9: it should be 61 instead of 60
Thank you for your accurate observation; we corrected it; it is indeed 61 since the increase is 60.90 times.
Comment 10: Why was the content of Asn and Gln not determined?
It is important to note that during the hydrolysis process for amino acid analysis, Asn and Gln tend to undergo deamination. This chemical alteration leads to their conversion into Asp (aspartic acid) and Glu (glutamic acid), respectively. As a result, Asn and Gln are indirectly measured by quantification of Asp and Glu. This information has been included in the manuscript to clarify the methodological approach and interpretation of our amino acid analysis results.
Comment 11: “table 4” ?
Thank you for your accurate observation; we corrected it in Table 3.
Comment 12: This fragment should be reworded because the results do not indicate what is described in the text.
We have added an appropriate paragraph in this Discussion section.
Comment 13: Are you sure that the standard deviations are calculated correctly? It is unlikely that they will be the same (in columns).
The error bars indicated in Table 3 are not standard deviations. They correspond to standard error. Furthermore, the standard error estimates may be similar due to the variability indicated for the average values of ORAC, ABTS, and DPPH adjusted by the Generalized Linear Model (GLM). In that sense, the reviewer indicates that it is unlikely to obtain similarity in the standard deviation bars (descriptive scatter bars), which is correct. However, our data correspond to standard error (inferential scatter bars), and due to the fit of the model, similar values can be obtained as a consequence of the goodness of fit of the GLM.
Comment 14: “In amaranth proteins” - What did you mean?
"In amaranth proteins" means the samples used from amaranth protein concentrate and digests. We have made this change in the manuscript.
Reviewer 3 Report
Comments and Suggestions for Authors
This study evaluated the impact of in vitro gastrointestinal digestion on protein digestibility, amino acid release, and antioxidant activity in proteins derived from amaranth (Amaranthus cruenthus L.) and cañihua (Chenopodium pallidicaule Aellen).The results suggested that both amaranth and cañihua proteins, after simulated gastrointestinal digestion, showed high digestibility, and released peptides and amino acids with potent antioxidant properties, underscoring their potential health benefits. This is an important contribution to the field of nutrition and antioxidant. I have a few comments, explained below. I hope that my comments are very useful for the improvement of this research.
Comments
(1) L28: AAC -> CAA?
(2) L72-73: Instead of indicating high protein content, please show the protein contents of amaranth and cañihua.
(3) L104: Please show the yield of protein extracted from amaranth and cañihua seed.
(4) L142: Why did the authors use the BCA method to determine protein digestibility? The BCA method is usually used as a protein quantification. What the authors want to know is the degree of protein digestibility. Generally, the digestibility would be determined by measuring the amount of amino acids using picrylsulfonic acid.
(5) HepG2 cell: Peptides with large molecular weights produced in the gastrointestinal tract are rarely transported directly to the liver. Please show the reason for using HepG2 cells in this study. In addition, the limitations of this experiment should be indicated.
(6) Fig.2: I don't think the authors need a border for the figure.
(7) Standard protein: In this study, the authors evaluated the antioxidant activity of the amaranth and cañihua seed digestion compared with Blank. It is not surprising to see antioxidant activities of the amaranth and cañihua seed digestion when compared to Blank. Is there a standard protein commonly used for such tests? If there is, I think it should be compared to the degradation products of that protein.
Author Response
Respuesta a los comentarios del revisor 3 ( 2711459 )
Agradecemos sus valiosas sugerencias y comentarios constructivos, que nos han ayudado a mejorar la calidad del manuscrito. Hemos incorporado sus sugerencias y comentarios en el manuscrito revisado. A continuación se presentan respuestas detalladas punto por punto a su revisión. Sus comentarios aparecen en texto negro, seguidos de nuestras respuestas punto por punto en azul.
Comentarios
(1) L28: AAC -> CAA?
Gracias por detectar esta inconsistencia. De hecho, "AAC" debería haber sido "CAA" (Actividad Antioxidante Celular). Ha sido corregido.
(2) L72-73: En lugar de indicar alto contenido de proteínas, muestre el contenido de proteínas del amaranto y cañihua.
Ahora hemos incluido los contenidos proteicos específicos del amaranto y cañihua en el manuscrito revisado. El amaranto y la cañihua tienen contenidos de proteína de aproximadamente 13,6% y 15,2%, respectivamente, en base a peso seco.
(3) L104: Favor indicar el rendimiento de proteína extraída de la semilla de amaranto y cañihua.
El rendimiento de proteína extraída de las semillas de amaranto y cañihua fue de aproximadamente 23% y 19%, respectivamente.
(4) L142: ¿Por qué los autores utilizaron el método BCA para determinar la digestibilidad de las proteínas? El método BCA se suele utilizar como cuantificación de proteínas. Lo que los autores quieren saber es el grado de digestibilidad de las proteínas. Generalmente, la digestibilidad se determinaría midiendo la cantidad de aminoácidos usando ácido picrilsulfónico.
Nos gustaría agradecerle mucho por este comentario. La digestibilidad in vitro de los concentrados proteicos se calculó mediante el método de Bradford. Ya realizamos la corrección en el manuscrito. Este método se basa en monitorear y determinar el contenido de proteína soluble que disminuyó gradualmente durante la digestión en comparación con el contenido de proteína soluble antes de la digestión. El método de Bradford funciona bien ya que el tinte azul de Coomassie se une a las proteínas, formando complejos proteína-tinte y exhibiendo una coloración azul. Además, se ha descrito que el método de unión del tinte azul de Coomassie no es sensible para cuantificar péptidos con un peso molecular inferior a 3 kDa porque los péptidos de cadena corta no formarán los complejos proteína-tinte que provocan el cambio de color. Por lo tanto, a medida que la proteína se degradaba gradualmente durante la digestión, la digestibilidad aumentaba en consecuencia.
(5) Célula HepG2: los péptidos con grandes pesos moleculares producidos en el tracto gastrointestinal rara vez se transportan directamente al hígado. Muestre el motivo del uso de células HepG2 en este estudio. Además, conviene indicar las limitaciones de este experimento.
Elegimos células HepG2, una línea celular de hígado humano, para nuestro análisis de antioxidantes en función de su origen hepático. El hígado desempeña un papel fundamental en el metabolismo de los xenobióticos, lo que lo convierte en un sitio fundamental para estudiar la respuesta metabólica a los compuestos bioactivos. Las células HepG2, bien diferenciadas y transformadas, proporcionan un modelo in vitro fiable ampliamente utilizado en análisis de antioxidantes e investigaciones bioquímicas y nutricionales. Sus características, similares a las del hígado humano, nos permiten simular el metabolismo del hígado humano y evaluar el impacto potencial de varios compuestos. Sin embargo, es importante reconocer las limitaciones de este enfoque in vitro. Si bien las células HepG2 proporcionan información valiosa, no pueden replicar completamente la complejidad de las funciones hepáticas in vivo y las interacciones con otros sistemas biológicos.
(6) Fig.2: No creo que los autores necesiten un borde para la figura.
Siguiendo su sugerencia, se eliminó el borde alrededor de la Figura 2 para mejorar la claridad visual y la coherencia con el resto de las figuras.
(7) Proteína estándar: En este estudio, los autores evaluaron la actividad antioxidante de la digestión de las semillas de amaranto y cañihua en comparación con Blank. No es sorprendente ver actividades antioxidantes en la digestión de las semillas de amaranto y cañihua en comparación con Blank. ¿Existe una proteína estándar que se utilice habitualmente para este tipo de pruebas? Si lo hay, creo que debería compararse con los productos de degradación de esa proteína.
En el manuscrito revisado, hemos incluido una comparación de las actividades antioxidantes de los digeridos de semillas de amaranto y cañihua con otras proteínas de diferentes fuentes vegetales para proporcionar un contexto más sólido a nuestros resultados. Para trabajos futuros, utilizamos un control positivo de las actividades antioxidantes utilizando un antioxidante conocido.
Apreciamos enormemente la oportunidad de mejorar nuestro manuscrito con sus valiosas contribuciones y confiamos en que estas modificaciones hayan mejorado la calidad y claridad de nuestro trabajo.

Round 2
Reviewer 1 Report
Comments and Suggestions for Authors
Introduction
Page 2, line 78 – Please consider replacing “Researching” with “The evaluation of…”
Page 2, line 79 - I suggest this alternative: “...strategies to promote their consumption and prevent chronic diseases.”
Page 2, line 80 – The release of peptides was not addressed in this study and therefore, I suggest deleting “of peptides”.
Results
Page 6, lines 268-274 – My previous comments on these sentences were just to calculate the protein content on a dry weight basis or on a dry weight basis and fibre-free basis in order to compare the protein content of different products.
Page 8, lines 324 and 327 – It is “A. cruentus species” because, as you know, the scientific name is given as its genus followed by the specific epithet.
Page 13, line 469 – In my previous comment I mentioned the quotation [61] instead of [51]. I apologize for that. In fact, I considered that the authors could include a short text to justify the reference [51]. Thus, I would suggest including after “antioxidant activity” the following words: “as reported by Gallego et al. [51]”, for instance.
Page 13, line 473 – I suggest including [53] after “respectively” in the previous sentence and before “However”.
Page 13, lines 500 and 501 – The authors agreed with the reviewer’s comment on these sentences and answered “The text has been modified...” but the text in this second version of the manuscript is the same of the first version.
Page 14, line 505 – Similarly, the authors’ answer says the manuscript was “corrected to read “...[25]. Major antioxidant...”, but the text of the new version of the manuscript was not modified.
Page 14, lines 507 and 508 – It is “mg TE mg-1”.
Page 14, lines 520 and 521 – Please consider this alternative: “...favors the ABTS scavenging activity”.
Page 15, Table 3, first line – It is “Protein concentrate”.
Page 15, lines 595-598 – It seems more adequate a sentence like this one: “Similarly, all fractions had no significantly different high CAA ranging from (the lowest value) in GD and 85.55 % in ID < 5 kDa for a concentration of 0.750 mg mL-1.”
Page 17, line 622 – The authors agreed to include “their” before “fractions” but it was not done in this version of the manuscript.
Comments on the Quality of English LanguageThe English Language is fairly good.
Author Response
Response to Reviewer 1’s Comments (2711459)
We thank you for your valuable suggestions and constructive comments. Your meticulous review has significantly enriched and improved the quality of our manuscript. We have incorporated your recommendations in the revised version of the manuscript, which are indicated in red text.
A detailed response to each of your comments is presented below.
Comments and Suggestions for Authors
Comment 1: Page 2, line 78 – Please consider replacing “Researching” with “The evaluation of…”
We appreciate your suggestion to improve clarity in line 78. We have replaced "Researching" with "The evaluation of".
Comment 2: Page 2, line 79 - I suggest this alternative: “...strategies to promote their consumption and prevent chronic diseases.”
We value your recommendation; we have incorporated the suggested phrase.
Comment 3: Page 2, line 80 – The release of peptides was not addressed in this study and therefore, I suggest deleting “of peptides”.
We have removed the phrase "of peptides" following your suggestion.
Comment 4: Page 6, lines 268-274 – My previous comments on these sentences were just to calculate the protein content on a dry weight basis or on a dry weight basis and fibre-free basis in order to compare the protein content of different products.
We appreciate your clarification regarding your previous comment and will consider the suggestion to remove the fiber in future analyses to facilitate the comparison of protein content between different products. We have reviewed the data and confirm that the protein content presented in the manuscript corresponds to the freeze-dried protein concentrates of both pseudocereals. We indicate this in the manuscript.
Comment 5: Page 8, lines 324 and 327 – It is “A. cruentus species” because, as you know, the scientific name is given as its genus followed by the specific epithet.
Thank you for pointing out this error. We have corrected the scientific name to 'A. cruentus species'.
Comment 6: Page 13, line 469 – In my previous comment I mentioned the quotation [61] instead of [51]. I apologize for that. In fact, I considered that the authors could include a short text to justify the reference [51]. Thus, I would suggest including after “antioxidant activity” the following words: “as reported by Gallego et al. [51]”, for instance.
We have modified these lines as per your suggestion. Thank you.
Comment 7: Page 13, line 473 – I suggest including [53] after “respectively” in the previous sentence and before “However”.
We have added the citation where suggested. We have also adjusted the numbering since the order of appearance corresponds to [52].
We adjusted the bibliography accordingly on Page 20, lines 799 and 803.
52. Montoya-Rodríguez, A.; González, E.; Díaz, V.; Reyes-Moreno, C.; Milán-Carrillo, J. Extrusion improved the anti-inflammatory effect of amaranth (Amaranthus hypochondriacus) hydrolysates in LPS-induced human THP-1 macrophage-like and mouse RAW 264.7 macrophages by preventing activation of NF-kB signaling. Mol. Nutr. Food Res. 2014, 58, 1028–1041. DOI: 10.1002/mnfr.201300764.
53 Vilcacundo, R.; Martínez-Villaluenga, C.; Miralles, B.; Hernández-Ledesma, B. Release of Multifunctional Peptides from Kiwicha (Amaranthus caudatus) Protein under In Vitro Gastrointestinal Digestion. J. Sci. Food Agric. 2019, 99(3), 1225–1232. DOI: 10.1002/jsfa.9294
Comment 8: Page 13, lines 500 and 501 – The authors agreed with the reviewer’s comment on these sentences and answered “The text has been modified...” but the text in this second version of the manuscript is the same of the first version.
We appreciate the observation and apologize for the error in updating the manuscript. We have now effectively modified these lines to reflect the changes suggested by your previous review.
Comment 9: Page 14, line 505 – Similarly, the authors’ answer says the manuscript was “corrected to read “...[25]. Major antioxidant...”, but the text of the new version of the manuscript was not modified.
Thank you for pointing out this omission. We have revised and corrected line 505 to correctly reflect the citation [25] and the phrase "Increased antioxidant activity" as per your recommendation.
Comment 10: Page 14, lines 507 and 508 – It is “mg TE mg-1”.
We have corrected the unit to 'mg TE mg-1'.
Comment 11: Page 14, lines 520 and 521 – Please consider this alternative: “...favors the ABTS scavenging activity”.
We have replaced the phrase as per your suggestion. Thank you.
Comment 12: Page 15, Table 3, first line – It is “Protein concentrate”.
We have corrected the first line of Table 3.
Comment 13: Page 15, lines 595-598 – It seems more adequate a sentence like this one: “Similarly, all fractions had no significantly different high CAA ranging from (the lowest value) in GD and 85.55 % in ID < 5 kDa for a concentration of 0.750 mg mL-1.”
We have modified these lines as per your suggestion. Thank you.
Comment 14: Page 17, line 622 – The authors agreed to include “their” before “fractions” but it was not done in this version of the manuscript.
We have added “their” before “fractions” in line 622.
Once again, we sincerely thank you for your dedication and time invested in reviewing our work. Your input has improved this manuscript.

Reviewer 2 Report
Comments and Suggestions for Authors
-
Author Response
We sincerely thank you for your dedication and time in reviewing our work. Your input has improved this manuscript.
Reviewer 3 Report
Comments and Suggestions for Authors
I am satisfied with the revisions that have been made by the authors.
Author Response

(The authors gave the same response as above.)
